# Hands Up Program: Results of a feasibility study of a randomized controlled trial of a bone health exercise and education program for adults aged 50–65 post distal radius fracture

**Christina Ziebart** [1] *, **Joy MacDermid** [1,2], **Dianne Bryant** [1], **Mike Szekeres** [1], **Nina Suh** [3]

**1** School of Physical Therapy, Department of Health and rehabilitation Sciences, University of Western Ontario, London, Ontario, Canada, **2** Clinical Research Lab, Roth McFarlane Hand and Upper Limb Centre, St. Joseph's Health Centre, London, Ontario, Canada, **3** Emory Health Care, Atlanta, Georgia, United States of America

* cziebart@uwo.ca

## Abstract

### Purpose

Distal radius fractures (DRF) that occur from a fall from level ground are considered fragility fractures and may be the first indication that someone has compromised bone mineral density and is at risk of having osteoporosis. Women at about 50 years of age experience a dramatic increase risk of these fractures. Rehabilitation after DRF focuses on restoring range of motion and strength in the wrist, but rarely address future bone health concerns. We developed a 6-week therapist-developed remote full body exercise and osteoporosis/bone-health education program (called The Hands Up Program). This study was designed to evaluate the feasibility of a larger trial examining the effects of a whole-body exercise and education program for people 50–65 after a DRF.

### Methods

Community dwelling individuals between the age of 50–65 with a radiographically confirmed DRF were recruited from the Roth| McFarlane Hand and Upper Limb Center in London, Ontario. Participants were randomized in a 1:1 ratio into either the Hands Up Program which was a twice weekly exercise and education program for 6 weeks, or the control group where they proceeded with usual care. The intervention was delivered online through a website where participants created a unique username and password to access. The primary feasibility outcomes of the study were recruitment rate (74 participants in 1 year), retention rate (75% completion), and intervention adherence rate (60% of completion of the exercise program). Secondary outcomes included strength, range of motion, self-reported outcomes, and bone density.

**Data Availability Statement:** All relevant data are within the manuscript and its Supporting Information files.

**Funding:** This study was funded in part by the Canadian Institutes of Health Research (CIHR) in the form of a Foundation Grant [167284] to JM.

**Competing interests:** The authors have declared that no competing interests exist.

## Results

Overall, 74 participants were recruited in 14 months. Retention did not meet the criteria for success, as only 53% of the participants attended their 12-month visit. Adherence was also not met albeit close with 55% completing the exercise program. Twelve participants withdrew from the study, five due to the time commitment, four without explanation, one due to group allocation, one due to COVID and one participant moved. One participant was deemed ineligible after consent. Four of the participants that withdrew were in the intervention group, and four in the control group, five participants withdrew before they were allocated to a group. Feedback from patients indicated potential improvements to the program: lower assessment burden, spacing out the intervention so that the education portion could be delivered during the immobilization phase of recovery, and creating a more individualized program.

## Conclusion

Adherence and retention were both a challenge, although adherence was close to target. While achieving adherence to exercise in bone health is a known challenge, given the importance of prevention in the at-risk DRF population and the challenges in co-design and delivery during the pandemic, we believe evaluation of a revitalized program is warranted.

## Introduction

Osteoporosis is a bone disease that occurs from a decrease in bone mineral density, increasing the risk of non-traumatic fractures. Fractures in the wrist, hip, and vertebrae are the most common. However, a fracture at the wrist is usually the first indication of the presence of osteoporosis [1]. There have been several factors reported to be associated with negative outcomes after a distal radius fracture (DRF), such as higher age [2], female sex [2] and joint involvement during play and work (i.e. jobs requiring high demands from the hand and wrist typically have worse outcomes than jobs that require minimal use of the hand and wrist) [3, 4] and are potentially related to sustained feelings of pain and disablement. A population study of more than 47,000 patient who incurred a DRF over a 4 year interval in one province in Canada found 73% were women with a rapid spike in incidence at age 50 [5]. It was hypothesized that the high incidence of fractures found in those people aged 51–65 is likely due to changes in bone mineral density (BMD), potentially caused by osteoporosis. The unique aspect of this fragility fracture is that it is typically the earliest in a compromised bone health trajectory and thus may be a signal that nonpharmacological management of bone health is both indicated and most likely to be effective. However it is also true that in the 50 to 65-year-old age group many people are still active occupationally and in sports which may also contribute to fractures [6]. This was confirmed in a cluster analysis study of 968 middle and older-aged adults from Canadian Longitudinal Study of Aging [7]. The patients were approximately evenly distributed between 2 clusters. The "misfortunate" cluster included a predominantly younger cohort which were physically active, with less comorbid conditions, better bone health, and better general health than the other cluster who exemplified fragility fractures as they were relatively compromised in these same areas [7].

People with osteoporosis who are not taught to modify their lifestyle, specifically through exercise, nutrition and fall prevention strategies, are more likely to suffer more debilitating

fractures of the hip and spine later in life [8]. Current wrist fracture rehabilitation focuses on restoration of joint mobility and hand function [1], and focuses on those over the age of 65. However, those 50–65 years of age represent the largest group of people presenting with wrist fractures [9]. They may require more targeted rehabilitation approaches since their bone health and activity levels are different from older adults. Further, early intervention provides the opportunity for preventing future debilitating fractures of the spine or hip.

There is little information conveniently accessible to people related to exercise, nutrition and fall prevention to reduce the risk of osteoporotic fractures [10]. Therefore we developed the Hands Up Program. The Hands Up Program targets individuals after they sustained a DRF and if they are between the age of 50–65. The intention of the program is to complement rehabilitation of the DRF with exercises and education that can address their risk of osteoporosis. The goal of this research is to develop and test a novel exercise and education group intervention (Hands-Up) that will incorporate education on bone health, prevention guidelines, nutrition and fall prevention, with an online exercise class and a structured virtual home safety assessment, which aligns with the osteoporosis guidelines [11]. The protocol has been previously published. The current study is addressing the feasibility of the program in patients 50–65 with a low-impact DRF.

This multimodal intervention will target the major risk factors for secondary fractures, to enhance functional recovery and health following a DRF, compared with usual physical therapy for wrist fractures.

## Methods

### Study design

The full protocol for this two-arm, 1:1 pilot single-blinded parallel RCT has been previously described (ClinicalTrials.gov registration NCT03997682) [12]. Briefly, Hands Up trial recruited community-dwelling adults between the age of 50–65 years with a radiographically-confirmed DRF. Participants were randomized through simple random sampling using opaque envelops, by the lead student investigator, into either the Hands Up Program where they performed an online home whole body exercise and education program, twice a week for 6 weeks, or into the control group where they proceeded with their usual care. The allocation did not have stratification or permuted blocks. Participants were recruited through the Roth| McFarlane Hand and Upper Limb Center (RM-HULC). Participants were identified through chart review and treating physicians. Recruitment began on March 18, 2021, and concluded on April 29, 2022.

Participants completed all outcome assessments at baseline, 6 weeks, 3 months, 6 months, and 12 months. The objective outcome measures (reported below) were administered by a blinded research assistant (RA) in a research lab at the RM-HULC, and self-reported outcome measures were conducted online at the participant's home, to reduce the amount of time spent in the hospital during the COVID-19 pandemic. Participants were instructed to complete daily diaries (brought in at each study visit) to record physical activity, falls and health-related events [29]. All participants gave their informed, written consent and the study was approved by the Western University research ethics board, in July 2019, reference number 2019-114095-27359. Continue ethics have been approved by the Western Research Ethics Board yearly (most recently in July 2024, reference number: 2024-114095-95039).

### Participants

Participants were included if they were a patient at the RM-HULC, if they had a DRF within the last 6–10 weeks, aged 50–65, were able to speak and understand English and were able to

provide informed consent. Participants were excluded if they had any contraindications to exercise, progressive neurological disorders that would affect study participation, unable to stand or walk independently and unable to provide consent.

## Intervention/control activities

The exercise and education program has been described in detail previously [12]. The exercise program targeted strengthening for the upper extremity, lower extremity, range of motion for the upper extremity and static and dynamic balance training. The intervention group participated in a 6-week exercise program with a progression of the exercises occurring at 3 weeks, which began immediately after their cast was removed. The first three weeks were focused on hand, wrist and shoulder rehabilitation and included lower extremity exercises and balancing. The intention was to support the rehabilitation post-DRF but also introduce exercises that would benefit compromised bone health, like osteoporosis. The lower extremity exercises progressed by adding weight or include a harder variation of the movement, and the upper extremity exercises focused on adding load through the fractured limb, for the intention of aligning the program with osteoporosis exercise recommendations. The control group participated in their usual care which would typically involve hand therapy for approximately 6 weeks after their cast has been removed.

## Outcome measures

The primary outcomes of the study reflected feasibility for a larger trial. Feasibility outcomes were the number of participants recruited (goal-74 in 12 months), the number of participants retained until study completion (goal-75% at 12 months), and the proportion of participants who adhered to Hands Up Program (goal -60% adherence). Additional outcome measures were assessed to gain insight into preliminary effects of the program. Secondary outcomes included gait speed, five times sit to stand, semi tandem stand, tandem stand, single leg stand, timed up and go, Biodex balance assessment, grip strength, pinch strength, wrist range of motion (pronation, supination, flexion, extension, radial deviation, ulnar deviation), hip and waist circumference, bone mineral density, EQ5DL, patient-rated wrist evaluation (PRWE), disability of the arm, shoulder and hand (DASH), global rating of change scale (GRC), osteoporosis quality of life questionnaire (OQLQ), fear of falling questionnaire, and personalized exercise questionnaire.

## Qualitative analysis

Participants in the intervention were also asked to participate in a semi-structed exit interview to discuss their experience in the program. The interviews were recorded and transcribed verbatim. The interviews were analyzed using interpretive descriptive methods [13] to identify key themes to improve the Hands-Up Program. Participants were prompted to discuss their general impression of the program, the exercise portion, the education portion, the outcome measures, whether they would continue the program and any additional thoughts.

## Statistical analysis

Recruitment rates were defined as the total number of participants randomized over the 12-month recruitment window. Retention was the count of participants that had any data at the data collection time point. No imputation of missing data was performed. One participant in the intervention group did not followed up and therefore there is missing data on adherence and retention, and it is assumed that zero sessions were completed. Pearson's R correlations

was conducted as a preliminary evaluation of the outcome measures included this study to explore redundant outcome measures and measures that show preliminary indication of an effect of the intervention. Correlations of 0.8 or greater which are typically deemed very strong or perfect is considered redundant and one of the outcomes will be removed. A correlation of 0.6–0.79 is considered strong, 0.4–0.59 is considered moderate, 0.2–0.39 is considered weak and <0.2 is considered very week or no relationship [14]. Statistically analyses were conducted in SPSS version 14.

## Results

### Participants characteristics

There were 272 adults aged 50–65 with a suspected DRF that were further screened for eligibility, all through the RM-HULC. Of these, 14 were not eligible to participate, 130 expressed interests in participating in the study, 74 consented to participate and 69 attended the baseline visit. Of those that were not eligible, 2 had a self-reported cognitive impairment or were chronically ill, 6 did not have a DRF, 1 participant did not meet the DRF timeline of being within 6–10 weeks, 4 did not have access to a computer, 1 could not speak or understand English, and did not have access to support for translation. Four participants withdrew their consent before attending their baseline visit. One participant withdrew their consent because of COVID and did not want to risk potential exposure, two withdrew without an explanation and one was not eligible (Fig 1). 69 participants were randomized between March of 2021 and May 2022, with 34 in the intervention group and 35 in the control group (Table 1). Twelve participants withdrew from the study, five due to the time commitment, four without explanation, one due to group allocation, one due to COVID and one participant moved. One participant was deemed ineligible after consent. Four of the participants that withdrew were in the intervention group, and four in the control group, five participants withdrew before they were allocated to a group. No participants who withdrew from the study opted to withdraw their data.

### Recruitment

We recruited 66 participants within the a priori timeline of 12-months and 74 in 14-months. Fig 2 demonstrates the number of participants recruited per month within the year. All recruitment took place at one site, the RM-HULC. The age restriction of this study reduced the number of participants eligible to participate. Additionally, participants travel from across Ontario to receive care, and many of the eligible participants declined to participate due to the commute. Participants were screened during surgeon clinic day sheets and through the urgent care center. Three of the participants were identified to have a DRF and consented to participate while they were still in their cast, and then did not return for their baseline visit and no further follow up could be achieved.

### Retention

Twelve participants (16%) withdrew from the study. The most common reason for withdrawal was due to the time commitment of the study (n = 4, 5%). Four (5%) of the participants withdrew without an explanation. Several of the participants did not attend all study visits or did not complete the online questionnaires at certain time points. A total of 69 (93%) participants completed the baseline visit, 55 (74%) participants completed the 6- week visit, 43 (58%) completed the 3-month visit, 45 (61%) completed the 6-month visit and 39 (53%) completed the 12-month visit. When adjusting for the number of participants that completed the baseline visit as the denominator instead of the number of participants that consented then 79%

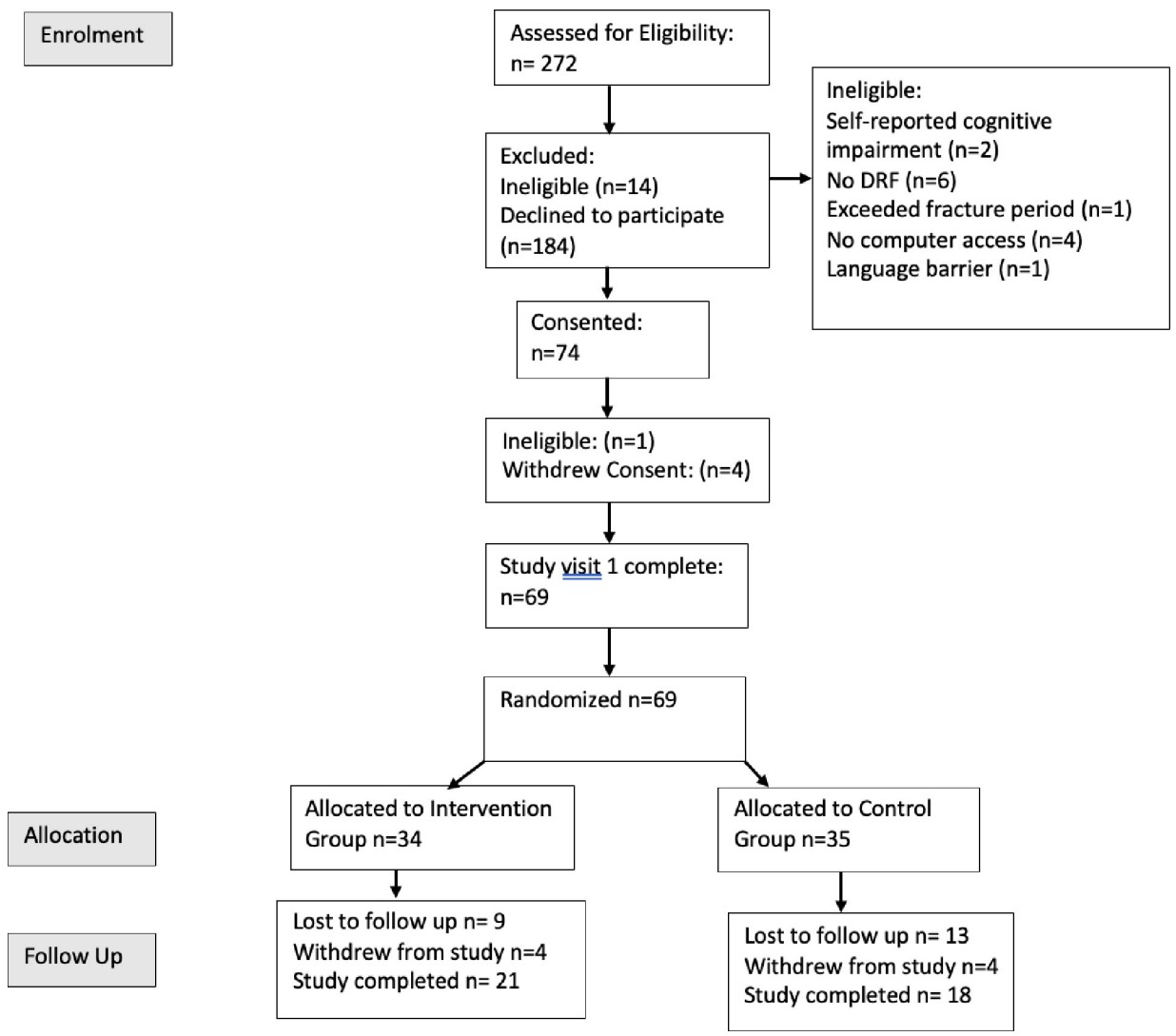

**Fig 1. Participant allocation and randomization from RCT.**

completed the 6-week visit, 62% completed the 3-month visit, 65% completed the 6-month visit and 57% completed the 12-month visit. The retention rate did not meet our prior criteria of 75%. As part of the exit interview, participants were asked to comment on the outcome measures. Many of the participants mentioned that there were too many questionnaires and completing the questionnaires was a significant time commitment. Most of the participant were comfortable attending RM-HULC for the physical assessments. However, as mentioned in recruitment, those unwilling to attend RM-HULC did not consent to participate.

## Adherence

Overall, adherence was 55%, therefore approaching but not meeting the a priori criterion of 60% adherence to the 6-week twice weekly exercise and education program. Eleven participants did not complete any of the recommended sessions, which may have been due to the online nature of the program, as two of the participants did not sign up for the website. Additionally, three participants missed several visits and did not responded to follow-up requests,

**Table 1. Baseline participant characteristics.**

| Measure | Entire Cohort n = 63 | Intervention n = 30 | Control n = 33 | Male n = 7 | Female n = 56 |
|---|---|---|---|---|---|
| Age (year): Mean (SD) | 60 (3.4) | 60.7 (3.0) | 60.1 (3.8) | 59.8 (4.0) | 60.5 (3.4) |
| Height (cm): Mean (SD) | 168.7 (8.7) | 167.9 (8.8) | 169.5 (8.7) | 184.6 (9.2) | 166.7 (6.2) |
| Weight (lbs): Mean (SD) | 168.0 (42.7) | 171.5 (39.4) | 165.0 (45.9) | 236.7 (57.5) | 159.0 (31.1) |
| Body mass index: Mean (SD) | 26.6 (5.6) | 27.6 (5.4) | 25.9 (5.7) | 31.7 (8.5) | 26.01 (4.8) |
| Waist Circumference (cm): Mean (SD) | 85.7 (19.3) | 84.4 (21.7) | 86.8 (17.2) | 88.9 (45.6) | 85.3 (15.5) |
| Hip Circumference (cm): Mean (SD) | 102.0 (20.1) | 100.9 (25.6) | 102.9 (14.0) | 112.3 (10.8) | 100.9 (20.5) |
| Diagnosed with osteoporosis: count | 42 | | | | |
| Side of fracture: count | Right: 42 Left: 21 | | | | |
| Dominant side fractured: count | 41 | | | | |
| Treatment: count | Splint: 43 Cast: 43 Surgery: 43 Other: 43 | | | | |

so it is assumed that zero sessions were completed. Two participants said they were too busy and therefore did not engage in any of the sessions. Fourteen of the participants completed all 12 sessions. If the three participants with missing data are removed the adherence rate changes to 60%, which would then meet the a priori criterion.

Feedback in the exit interview of those in the intervention group revealed that participants generally enjoyed the program. However, there were several people that did not complete the exercise sessions because they wanted to return to their pre-fracture activities. One participant explained that she did not prefer to exercise inside, in front of her computer, and instead preferred physical activity by walking outside and working on her farm. Some participants felt that the intervention was too easy or wanted variability in the videos. Other participants said that the exercises were a good way to get back into an exercise routine. One participant

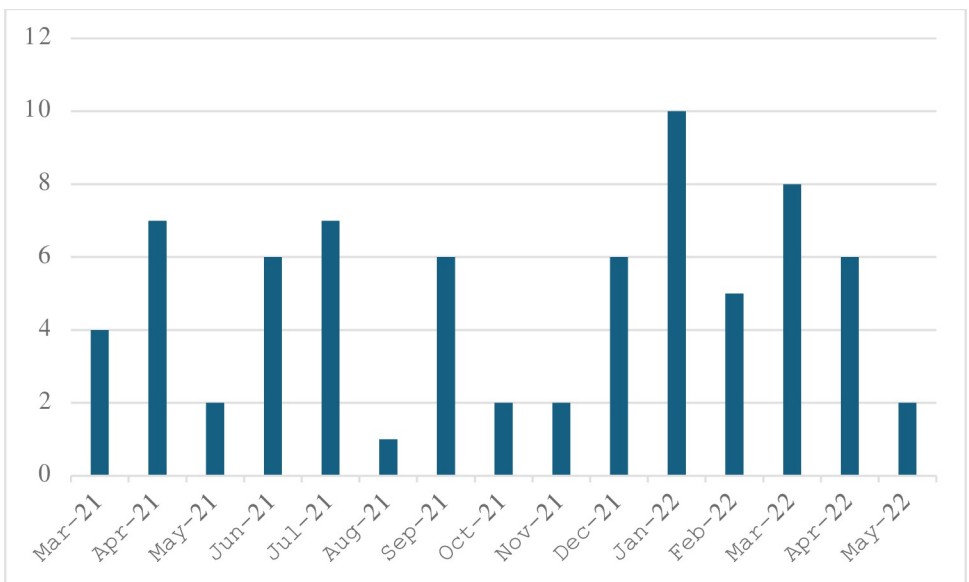

**Fig 2. Number of participants consenting per month in the 14-month recruitment period.**

expressed that they used the intervention as motivation to start walking again and engage in more daily activities. Several participants mentioned that there was a lot of information in the education program, and one participant, suggested giving the education modules during the immobility phase of the distal radius fracture to balance the time commitment. When prompted about the information in the education videos, participants said that it was thorough and that they enjoyed the information.

Adverse events were captured through the online questionnaires and direct reporting from the participants. Seven participants mentioned injuries throughout the participation, none of which were related to the intervention. Of the seven participants, three were in the control group and four were in the intervention group. Two of the participants tripped outside and re-fractured their distal radius, one participant underwent carpal tunnel surgery shortly after enrolling the program, one participant fell outside but it did not result in a fracture, another participant had a torn rotator cuff (she was in the control group, therefore, unrelated to the intervention), one participant contracted COVID, and another participant experienced a pinched nerve in her back (she was also in the control group, therefore, unrelated to the intervention).

## Outcomes

Many secondary outcome measures were included in this study. Outcome measure burden was described as one of the key reasons participants withdrew from the study. Therefore, it will be necessary to reduce the number of outcomes in a future larger trial. When looking at the Pearson's Correlation Coefficients, the following outcomes has a strong correlation but none of the outcomes had a very strong correlation: TUG and 5 times sit to stand (0.651), wrist flexion on the affected side and ulnar deviation on the affected side (0.626), wrist flexion on the affected side and radial deviation on the affected side (0.601), wrist flexion on the affected side and grip strength on the affected side (0.673), wrist extension on the affected side and ulnar deviation on the affected side (0.628), wrist extension on the affected side and radial deviation on the affected side (0.661), wrist extension on the affected side and grip strength on the affected side (0.641), ulnar deviation on the affected side and grip strength on the affected side (0.745), and radial deviation on the affected side and grip strength on the affected side (0.657).

The correlation table is uploaded as S1 File.

## Consideration for future trial

A fully powered RCT with a population sample of approximately 80,000 [5] and a 95% confidence interval with a 5% margin of error would be approximately 380 participants. Our recruitment target was achieved in 14 months at this single site, with 272 participants that would have been eligible at the single site. Therefore, a larger trial at the single site might be accomplished in 5 years with the recruitment rate experienced in this study. To recruit more participants, quicker, to achieve the desired sample size in a larger trial, more sites could be included.

## Discussion

This feasibility trial provides insight into the recruitment, retention and adherence of adults aged 50–65 with a DRF and the use of a whole body, home exercise and education program. Recruitment of this population proved to be successful, meeting the a priori criterion established. A fully powered RCT with grip strength as the primary outcome [15] would require a sample size of approximately 350 participants. This sample size could be achieved by including

more sites. Adherence and retention did not meet the a priori criteria but in some areas was close to targets which when considering patient feedback, low cost of the intervention, and the pandemic context means that failing to meet some targets was not a clear indication of lack of feasibility.

Recruitment was judged to successful, recruiting 66 of the 74 participants in 12 months (89%), and recruiting the additional 8 in the next two months. The most common reasons participants did not want to participant were due to the commute and the time commitment. Participants were asked to come into RM-HULC for the objective outcome measures, whereas the self-reported measures took place online at home. It seemed that the time commitment, largely related to the commute, was the biggest barrier to engaging in the program, which is consistent with barriers to exercise participation that has been previously cited [10, 16–18]. Adding additional sites is a strategy to achieve our recruitment for a future larger trial. Additional sites can also increase uptake of the program to reduce travel time for participants living outside of the London region.

Based on a preliminary evaluation of the outcome measures, we suggest reducing the number of outcome measures to reduce the participation burden on the participants, and hopefully reduce the time commitment for the study. During the study we tried to manage the time commitment and in-person commitment recognizing a potential fear of attending at hospital during the COVID-19 pandemic. If participants were interested in the program but had trouble attending the in-person visits, they were given the option to skip the in-person components. One participant accepted this offer but withdrew within the first 6 weeks. At the 12-month time point eight participants only completed the online portion and did not attend the in-person visit. However, there were also nine participants that only completed the in-person visits and did not complete the online outcome assessments at the 12-month visit. As virtual assessments become more sophisticated, it may not be necessary to have in-person assessment of performance in strength, balance and range of motion. Further exploration needs to be completed to determine if this was due to the time commitment or the participant's discomfort with using an online system. Four participants withdrew because of the time commitment of the program and the outcome measures so future iterations of the program should keep only the most pertinent outcome measures to reduce patient burden. One strategy might be to reduce the number of outcome measures at each time point, only asking certain questionnaires at certain time points. Also, with a younger population a critical look at whether there are ceiling or floor effects on some of the measures will also help to determine which outcome measures might be removed. Finally, we conducted correlations to identify redundant tests and grip strength was strongly correlated with the range of motion assessments. Grip strength should remain as an objective measure and subjective assessments can be used to evaluate function, with range of motion assessments removed in future iterations. As well, in a future iteration of this study, it would be recommended to provide the intervention group with the education materials while they are immobilized in their cast, and then begin the exercise intervention once their cast is removed. This would help to disperse the commitment of the study and help to balance the time commitment. Finally, some of the participants were lost to follow up. This study was also a large time burden on research staff, so future studies should consider using more automated systems or artificial intelligence to reduce research staff burden and hopefully reduce the number of participants lost to follow up.

Adherence to this program was low, and did not meet the a priori criterion; however, this is not unlike other exercise interventions [19–22]. Adherence to a home exercise program can be hard to measure, this study used a hard-copy exercise tracking sheet, but other methods have been cited in the literature [19, 20, 23, 24]. Participants were asked to engage in the Hands Up Program for 6-weeks, which was consistent with other interventions for people recovering

from a DRF [19, 20, 23–25], but it may have benefitted people with osteoporosis to engage in the whole body exercise portion of the program for up to 12 weeks to notice strength improvements, or up to one year, as was previously reported [26]. However, adherence declines with participants over time [26], so an intervention longer than 6 weeks may have resulted in an even worse adherence rate. Further this study did not evaluate the adherence of the usual group to their usual care activities, so it is not clear whether the adherence was comparable between the two groups.

This study was modified from its initial conceptualization, which was going to be an in-person exercise and education program, and instead became an online home exercise program because of the COVID-19 pandemic. Many participants chose not to participate in the study due to the commute, so it was likely a serendipitous change to improve recruitment and retention in the program. However, in person exercise programs have a better adherence rate than those that are online [27–30]. This program was intended to be pragmatic, encouraging participants to take ownership of their home exercise program, like that in an outpatient physiotherapy practice. To improve adherence and understanding of the exercises it may be helpful to have a physiotherapist check in with the participants after the first week, via a video call, to ensure the activities were properly conducted. This would be especially important if a future trial increases the age range of the participants, as the older participants are more likely to have compromised BMD and therefore increase their fracture risk [26]. Tutorial videos on how to do the exercises were created to help participants learn the exercise, but it might have been helpful to have a physiotherapist also check in. As well, to help address the time commitment of engaging in the program it might be helpful to have more, shorter, education videos. The program was designed to have 12 education videos, to mimic what an in-person education session might be. However, with the online format, many shorter, videos covering a single topic in 3 minutes or less might be more digestible and therefore increase adherence. Finally, when prescribing exercise, it may be necessary to adapt a more individualist approach, which may also help to improve adherence. An article on congenital heart defects begins to explore a more individualistic approach to exercise prescription [31], which may also be applicable for people with osteoporosis. The article on heart defects suggests starting the intervention with a history and physical exam, then assess based on a few key parameters, the provide recommendations for the type of exercise and intensity and finally follow up [31]. People with osteoporosis have a variety of exercise preferences [16], as well, some people found the Hands Up program challenging to adhere to, because they were already engaging in other activities. It might be helpful, when conducting the full-RCT for the Hands Up Program to consider conducting a more thorough history and physical exam and get a better sense of the exercise preferences. The program should ensure that all the participants are meeting the osteoporosis physical activity guidelines by conducting strength training twice weekly, balance training daily and finally at least 150 minutes of moderate-to-vigorous physical activity. Rather than requiring all the participants are engage in all the videos, we could prescribe specific videos to fill the gaps in their current fitness plan. For example, one of the participants really enjoyed cycling and walking, but did not participate in as much strength training. For this participant it would be helpful to emphasize the importance of the strength training program, to ensure adherence to the osteoporosis physical activity guidelines. Further consideration should be made for what the participants are doing for both exercise and physical activity. For example, one participant mentioned that she does a lot of farm work. Probing further into this would likely reveal that she is doing quite a bit of heavy lifting while also doing a lot of walking. For this participant, it might have been more valuable to emphasize the balance training and have a conversation about the intensity of the exercises to make sure the participant is always being challenged.

## Strengths and limitations

A strength of this study is that the intervention was created by the using expert input, evidence and qualitative information from patients. We enrolled both males and females to plan for future sex/gender-based analysis. We acknowledge several limitations to this study. Participants self-reported adverse events and therefore it's not clear whether the intervention resulted in any injuries. We used participant self-report to evaluate adherence which means adherence could be over-estimated, or under-estimated as three participants were assumed to have not engaged as they did not return their exercise tracker. Participants were not blinded to group allocation, to allow for the outcome assessor to be blinded, it required additional staff to do follow up visits and assess outcome. There is also a risk of contamination as those in the control may have sought out additional resources for exercise and education for bone health. Finally, we did not track adherence for the control group.

## Conclusion

In conclusion, this feasibility trial was satisfactory in recruitment, but not in adherence or retention. Suggestions for future trials include reducing the number of outcome measures, providing the intervention group with the education materials during their immobilization period, monitoring adverse events more closely, consider expanding the age criteria to 50 + and involving more than one recruitment site.

## Supporting information

**S1 Checklist. CONSORT 2010 checklist of information to include when reporting a randomised trial\*.**
(DOC)

**S1 File.**
(XLSX)

**S2 File.**
(XLSX)

**S3 File.**
(PDF)

## Acknowledgments

Joy MacDermid was supported by a Canada Research Chair in Musculoskeletal Health Outcomes and Knowledge Translation and the Dr. James Roth Chair in Musculoskeletal Measurement and Knowledge Translation. For the remaining authors, none were declared. Special acknowledgement to Katrina Munro, Rochelle Furtado, Steve Lu, Erfan Shafiee for their support in the recruitment and data collection throughout this project.

## Author Contributions

**Conceptualization:** Christina Ziebart, Joy MacDermid, Dianne Bryant, Mike Szekeres, Nina Suh.

**Data curation:** Christina Ziebart.

**Formal analysis:** Christina Ziebart.

**Funding acquisition:** Christina Ziebart, Joy MacDermid.

**Investigation:** Christina Ziebart, Joy MacDermid, Dianne Bryant.

**Methodology:** Christina Ziebart, Joy MacDermid, Dianne Bryant, Mike Szekeres, Nina Suh.

**Project administration:** Christina Ziebart.

**Resources:** Joy MacDermid, Mike Szekeres, Nina Suh.

**Software:** Joy MacDermid.

**Supervision:** Joy MacDermid, Dianne Bryant, Mike Szekeres, Nina Suh.

**Validation:** Christina Ziebart, Joy MacDermid.

**Visualization:** Christina Ziebart, Nina Suh.

**Writing – original draft:** Christina Ziebart.

**Writing – review & editing:** Christina Ziebart, Joy MacDermid, Dianne Bryant, Mike Szekeres, Nina Suh.

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
