## [Decision Letter · Decision Letter 0]

30 Aug 2024

PONE-D-24-19994Hands Up Program: results of a feasibility study of a randomized controlled trial of a bone health exercise and education program for adults aged 50-65 post distal radius fracturePLOS ONE

Dear Dr. Ziebart,

Thank you for submitting your manuscript to PLOS ONE. After careful consideration, we feel that it has merit but does not fully meet PLOS ONE’s publication criteria as it currently stands. Therefore, we invite you to submit a revised version of the manuscript that addresses the points raised during the review process.

We look forward to receiving your revised manuscript.

Kind regards,

Enock Madalitso Chisati, PhD

Academic Editor

PLOS ONE

Journal Requirements:

Reviewers' comments:

Reviewer's Responses to Questions

**Comments to the Author**

1. Is the manuscript technically sound, and do the data support the conclusions?

Reviewer #1: Yes

Reviewer #2: Partly

2. Has the statistical analysis been performed appropriately and rigorously? 

Reviewer #1: Yes

Reviewer #2: N/A

3. Have the authors made all data underlying the findings in their manuscript fully available?

Reviewer #1: Yes

Reviewer #2: Yes

4. Is the manuscript presented in an intelligible fashion and written in standard English?

Reviewer #1: Yes

Reviewer #2: Yes

5. Review Comments to the Author

Reviewer #1: This is an analysis of a carefully conducted pilot study to assess the feasibility of an exercise based intervention in a full-scale clinical trial. The feasibility study is well-designed, but the report is terse, and some of the details in the original published protocol referenced could be repeated. The manuscript is written in a conversational style that is sometimes off-putting: a patient "got COVID" (contracted), patients "did not mind", "to provide sufficient data to provide sufficient data", etc. A good editing with some more care to sentence structure and word choice is warranted.

One of the primary goals of a feasibility study is to provide information on the required sample size, which is given in the Discussion. This needs to be expanded to discuss the primary outcome, the hypothesis being tested, the test that will be used to test the hypothesis, and the assumptions that will be made gleaned from the feasibility study. Certainly attrition, retention, and adherence should be factored into this decision.

The randomization procedure used must be specified under CONSORT guidelines. 1:1 is not a randomization procedure. Permuted blocks? Complete randomization?

You talk about correlated outcomes and how you might use the observed correlation to fine tune your decisions for outcomes. This needs to be talked about here. In fact, you don't discuss what types of outcomes would be appropriate for the full-scale trial; would you just have one primary outcome? What hypotheses have you decided would be of interest?

In summary, a subsection of the results addressing statistical considerations of sample size and outcomes in the full-scale trial based on the feasibility trial would be a useful addition to this paper.

Reviewer #2: The manuscript examines the feasibility of an exercise and educational intervention for patients 50-65 years of age, presenting with a DRF. The manuscript concluded that such a programme is feasible and should continue onto a larger trial.

I think there is merit to what has been done (the amount of work that has gone into the study is unquestionable) and I do not necessarily disagree with the conclusion. I do feel, however, that the manuscript lacks certain information that is essential in fully justifying the conclusions. I provide the key points below, with some explanation and I think consideration and addressing teh comments will improve the manuscript.

Introduction:

There is little information about the hands-up programme, whether in the manuscript or the study's registration. Some more information (it could even be in the Methods), would provide a sense of what the requirements are, what it does exactly, how it would be structured if not affected by Covid-19. This will provide context to the reader for the challenges described, the solutions etc. It would also help clarify apparent discrepancies - e.g. p5 Lines 92-93 and p5 Lines 99-100, are these the same things, are both being targeted by the intervention?

Methods:

There needs to be a statement re the study having received Ethical approval.

Heads-up - as above, if the Introduction is not the place to describe the intervention, then it is in Methods.

Outcome measures - the outcome measures described in the manuscript, are 'half' of the ones described in the registration. Indeed, some physical measurements would help with the design of the larger trial, if nothing else to provide a sense of effect expected. Questionnaires were discussed, but not reported here. Interviews are reported, but not mentioned here. Perhaps linking with the 'more information for the intervention' comment, the rationale for the assessments should also be provided. For example, the measurements reported in the registration (e.g. handgrip strength, bone density) do not seem to align with the programme focusing on balance. Overall, the Methods make for an incomplete feasibility trial, while the Results appear to suggest that the necessary aspects were, in fact, present.

Results:

There were no results presented for the usual care group e.g. adherence, follow-up etc. It is hard to assess whether the Hands-up programme was eg un/feasible purely on the thresholds set, without comparing the findings to what happens with usual care. If, for example, usual care follow-up is 40% and hands-up is 60% that appears better than usual care, even if the threshold of e.g. 70% is not achieved.

Discussion:

Partially affected by the lack of information on the intervention, the recommendations made are not supported. Conjectures are presented frequently, as early as the Results ('..so other injuries may have occurred and were not reported..'). Several recommendations appear to have no basis for the suggested change; would that not end up becoming another feasibility trial? The evidence from the interviews would be useful in a more robust presentation regarding the outcomes, which would inform the subsequent trial.

Several discussion points appear to be contradicting, for example a) Covid-10 forced to online delivery which was not ideal as in-person delivery would be better vs an issue was the time commitment and travelling to the hospital (which would increase with full in-person delivery), b) increase the age bracket for better recruitment vs the actual aim of the study as described in the Introduction.

More clarity is needed throughout the section, for example a) novel exercise and education group intervention including education, prevention and nutrition vs enhancing functional recovery and health vs strength as an assessment criterion; how do these align.

Was the time commitment to the travel, the completion of the questionnaires or both the barrier (P13, Lines 274-276)?

Overall, I think there is merit in the study. In my view, however, it requires considerable revision to address the above issues, before it can be recommended for publication.

6. PLOS authors have the option to publish the peer review history of their article (what does this mean?). If published, this will include your full peer review and any attached files.

Reviewer #1: No

Reviewer #2: No

---

## [Author Response · Author response to Decision Letter 0]

16 Sep 2024

Reviewer #1: This is an analysis of a carefully conducted pilot study to assess the feasibility of an exercise based intervention in a full-scale clinical trial. The feasibility study is well-designed, but the report is terse, and some of the details in the original published protocol referenced could be repeated. The manuscript is written in a conversational style that is sometimes off-putting: a patient "got COVID" (contracted), patients "did not mind", "to provide sufficient data to provide sufficient data", etc. A good editing with some more care to sentence structure and word choice is warranted.

Thank you for this point. The manuscript has been reviewed and the language has been edited. 

One of the primary goals of a feasibility study is to provide information on the required sample size, which is given in the Discussion. This needs to be expanded to discuss the primary outcome, the hypothesis being tested, the test that will be used to test the hypothesis, and the assumptions that will be made gleaned from the feasibility study. Certainly attrition, retention, and adherence should be factored into this decision.

This is a great point. Thank you. I have included more depth in the discussion. 

The randomization procedure used must be specified under CONSORT guidelines. 1:1 is not a randomization procedure. Permuted blocks? Complete randomization?

Thank you. I have added more information: 

“Participants were randomized through simple random sampling using opaque envelops, by the lead student investigator, into either the Hands Up Program where they performed an online home whole body exercise and education program, twice a week for 6 weeks, or into the control group where they proceeded with their usual care. The allocation ratio was 1:1, with no stratification or permuted blocks”

You talk about correlated outcomes and how you might use the observed correlation to fine tune your decisions for outcomes. This needs to be talked about here. In fact, you don't discuss what types of outcomes would be appropriate for the full-scale trial; would you just have one primary outcome? What hypotheses have you decided would be of interest?

Thank you. I have included that in the discussion and results. I have also added more information about outcome measures in the methods: 

“Additional outcome measures were assessed to gain insight into preliminary effects of the program. Secondary outcomes included gait speed, five times sit to stand, timed up and go, Biodex balance assessment, grip strength, pinch strength, wrist range of motion (pronation, supination, flexion, and extension), hip and waist circumference, bone mineral density, EQ5DL, patient-rated wrist evaluation (PRWE), disability of the arm, shoulder and hand (DASH), global rating of change scale (GRC), osteoporosis quality of life questionnaire (OQLQ), fear of falling questionnaire, and personalized exercise questionnaire. Participants in the intervention were also asked to participate in an exit interview to discuss their experience in the program.”

In results: 

Consideration for Future Trial 

A fully powered RCT with grip strength as the primary outcome(20) would require a sample size of approximately 350 participants. Our recruitment target was achieved in 14 months at this single site, with 272 participants that would have been eligible at the single site. Therefore, a larger trial at the single site might be accomplished in 5 years with the recruitment rate experienced in this study. To recruit more participants, quicker, to achieve the desired sample size in a larger trial, more sites could be included or we could increase the age criterion to all participants over the age of 50, or through potentially modifying the time commitment associated with engaging in the study. 

Outcomes 

Many secondary outcome measures were included in this study. Outcome measure burden was described as one of the key reasons participants withdrew from the study. Therefore it will be necessary to reduce the number of outcomes in a future larger trial. When looking at the Pearson’s Correlation Coefficients, the following outcomes has a strong correlation and therefore may be redundant: TUG and 5 times sit to stand (0.651), wrist flexion on the affected side and ulnar deviation on the affected side (0.626), wrist flexion on the affected side and radial deviation on the affected side (0.601), wrist flexion on the affected side and grip strength on the affected side (0.673), wrist extension on the affected side and ulnar deviation on the affected side (0.628), wrist extension on the affected side and radial deviation on the affected side (0.661), wrist extension on the affected side and grip strength on the affected side (0.641), ulnar deviation on the affected side and grip strength on the affected side (0.745), and radial deviation on the affected side and grip strength on the affected side (0.657). 

The correlation table is uploaded as supplementary material. 

In summary, a subsection of the results addressing statistical considerations of sample size and outcomes in the full-scale trial based on the feasibility trial would be a useful addition to this paper.

Thank you. I have added this subsection to the results. 

Reviewer #2: The manuscript examines the feasibility of an exercise and educational intervention for patients 50-65 years of age, presenting with a DRF. The manuscript concluded that such a programme is feasible and should continue onto a larger trial.

I think there is merit to what has been done (the amount of work that has gone into the study is unquestionable) and I do not necessarily disagree with the conclusion. I do feel, however, that the manuscript lacks certain information that is essential in fully justifying the conclusions. I provide the key points below, with some explanation and I think consideration and addressing teh comments will improve the manuscript.

Introduction:

There is little information about the hands-up programme, whether in the manuscript or the study's registration. Some more information (it could even be in the Methods), would provide a sense of what the requirements are, what it does exactly, how it would be structured if not affected by Covid-19. This will provide context to the reader for the challenges described, the solutions etc. It would also help clarify apparent discrepancies - e.g. p5 Lines 92-93 and p5 Lines 99-100, are these the same things, are both being targeted by the intervention?

Thank you for this comment I have added more information to the introduction: 

“There is little information conveniently accessible to people related to exercise, nutrition and fall prevention to reduce the risk of osteoporotic fractures.(10) Therefore we developed the Hands Up Program. The Hands Up Program targets individuals after they sustained a DRF and if they are between the age of 50-65. The intention of the program is compliment rehabilitation of the DRF with exercises and education that can address their risk of osteoporosis. The goal of this research is to develop and test a novel exercise and education group intervention (Hands-Up) that will incorporate education on bone health, prevention guidelines, nutrition and fall prevention, with an online exercise class and a structured virtual home safety assessment. The protocol has been previously published. The current study is addressing the feasibility of the program. Specifically, is the Hands-Up program a feasible intervention for patients aged 50–65 with a low-impact DRF? 

This multimodal intervention will target the major risk factors for secondary fractures, to enhance functional recovery and health following a DRF, compared with usual physical therapy for wrist fractures.”

Methods:

There needs to be a statement re the study having received Ethical approval.

Sorry for that oversight. A statement on ethics has been added. 

Heads-up - as above, if the Introduction is not the place to describe the intervention, then it is in Methods.

Thank you. I have added it to the introduction since the protocol has already been published. 

Outcome measures - the outcome measures described in the manuscript, are 'half' of the ones described in the registration. Indeed, some physical measurements would help with the design of the larger trial, if nothing else to provide a sense of effect expected. Questionnaires were discussed, but not reported here. Interviews are reported, but not mentioned here. Perhaps linking with the 'more information for the intervention' comment, the rationale for the assessments should also be provided. For example, the measurements reported in the registration (e.g. handgrip strength, bone density) do not seem to align with the programme focusing on balance. Overall, the Methods make for an incomplete feasibility trial, while the Results appear to suggest that the necessary aspects were, in fact, present.

Thanks so much for this point.

I have added the following paragraph: 

‘Additional outcome measures were assessed to gain insight into preliminary effects of the program. Secondary outcomes included gait speed, five times sit to stand, timed up and go, Biodex balance assessment, grip strength, pinch strength, wrist range of motion (pronation, supination, flexion, and extension), hip and waist circumference, bone mineral density, EQ5DL, patient-rated wrist evaluation (PRWE), disability of the arm, shoulder and hand (DASH), global rating of change scale (GRC), osteoporosis quality of life questionnaire (OQLQ), fear of falling questionnaire, and personalized exercise questionnaire. Participants in the intervention were also asked to participate in a semi-structed exit interview to discuss their experience in the program. The interviews were recorded and transcribed verbatim. The interviews were analyzed using interpretive descriptive methods(13) to identify key themes to improve the Hands-Up Program. Participants were prompted to discuss their general impression of the program, the exercise portion, the education portion, the outcome measures, whether they would continue the program and any additional thoughts.’

Results:

There were no results presented for the usual care group e.g. adherence, follow-up etc. It is hard to assess whether the Hands-up programme was eg un/feasible purely on the thresholds set, without comparing the findings to what happens with usual care. If, for example, usual care follow-up is 40% and hands-up is 60% that appears better than usual care, even if the threshold of e.g. 70% is not achieved.

This is a really great point and something that we could do a better job of tracking in the larger trial. I’ll mention it in the discussion and limitations. 

Discussion:

Partially affected by the lack of information on the intervention, the recommendations made are not supported. Conjectures are presented frequently, as early as the Results ('..so other injuries may have occurred and were not reported..'). Several recommendations appear to have no basis for the suggested change; would that not end up becoming another feasibility trial? The evidence from the interviews would be useful in a more robust presentation regarding the outcomes, which would inform the subsequent trial.

Several discussion points appear to be contradicting, for example a) Covid-10 forced to online delivery which was not ideal as in-person delivery would be better vs an issue was the time commitment and travelling to the hospital (which would increase with full in-person delivery), b) increase the age bracket for better recruitment vs the actual aim of the study as described in the Introduction.

More clarity is needed throughout the section, for example a) novel exercise and education group intervention including education, prevention and nutrition vs enhancing functional recovery and health vs strength as an assessment criterion; how do these align.

Was the time commitment to the travel, the completion of the questionnaires or both the barrier (P13, Lines 274-276)?

Thank you for these thoughtful comments. I have done my best to address them throughout the discussion. 

Overall, I think there is merit in the study. In my view, however, it requires considerable revision to address the above issues, before it can be recommended for publication.

---

## [Decision Letter · Decision Letter 1]

29 Sep 2024

PONE-D-24-19994R1Hands Up Program: results of a feasibility study of a randomized controlled trial of a bone health exercise and education program for adults aged 50-65 post distal radius fracturePLOS ONE

Dear Dr. Ziebart,

Thank you for submitting your manuscript to PLOS ONE. After careful consideration, we feel that it has merit but does not fully meet PLOS ONE’s publication criteria as it currently stands. Therefore, we invite you to submit a revised version of the manuscript that addresses the points raised during the review process.

We look forward to receiving your revised manuscript.

Kind regards,

Enock Madalitso Chisati, PhD

Academic Editor

PLOS ONE

Reviewers' comments:

Reviewer's Responses to Questions

**Comments to the Author**

1. If the authors have adequately addressed your comments raised in a previous round of review and you feel that this manuscript is now acceptable for publication, you may indicate that here to bypass the “Comments to the Author” section, enter your conflict of interest statement in the “Confidential to Editor” section, and submit your "Accept" recommendation.

Reviewer #1: All comments have been addressed

Reviewer #2: (No Response)

2. Is the manuscript technically sound, and do the data support the conclusions?

Reviewer #1: (No Response)

Reviewer #2: Partly

3. Has the statistical analysis been performed appropriately and rigorously? 

Reviewer #1: (No Response)

Reviewer #2: No

4. Have the authors made all data underlying the findings in their manuscript fully available?

Reviewer #1: (No Response)

Reviewer #2: Yes

5. Is the manuscript presented in an intelligible fashion and written in standard English?

Reviewer #1: (No Response)

Reviewer #2: Yes

6. Review Comments to the Author

Reviewer #1: (No Response)

Reviewer #2: The authors have attempted to address my previous comments; this was successful at some points, while less so in others. I will try and discuss further the remaining points, with the aim to provide more clarity on the issues I think require attention. The more key ones are presented first.

Discrepancies throughout: there are still discrepancies in the manuscript. The selection of the age range is based on the thought processes, evidenced in the Introduction. To then recommend increasing the age range or stating that the 'age restriction.. reduced the.. participants', questions the rationale for those decisions. Further, increasing the age range potentially induces different issues (assuming an age-based digital divide, if the age range is increased to younger people, the difficulty in online assessment may be less, while if it increased to older people, the difficulty in online assessment may be more; transportation maybe less / more of a problem; time availability might be more / less of a problem (e.g. if >65 are retired) etc). Therefore, the solution can not be to alter parameters that would affect the process described.

Aim of the study: the aim of the programme is described as 'to complement rehabilitation of the DRF'. Its possible implementation is not clear. The description appears to present is as a standard addition to the usual care, while it appears it could be an option. In that respect, will it work for those who choose that option? Why is there insistence on a single site when travelling was reported as an issue, and delivering online at a different site might increase the uptake (close enough for the in-person visits, and online so they are reduced). There seems to be a need for a clearer 'so what' in the study.

Recommendations: There are several recommendations that seem unsupported and would alter the parameters of the feasibility trial. Please revisit and address accordingly. Perhaps a helpful starting point that would help with all of the above is Lancaster and Thabane (https://pilotfeasibilitystudies.biomedcentral.com/articles/10.1186/s40814-019-0499-1). This would help with providing a clearer direction.

Specific comments

Page 5, Line 95: please replace 'is compliment rehabilitation' with 'is to complement rehabilitation'

Page 5, Line 102: Please convert the question to statement.

Page 6, Line 116 - the ratio has already been reported earlier

Page 6, Line 120 and Page 6, Line 128: Please keep one of the two statements re Ethics. Also add the Body that provided the Ethical approval

Page 6, Line 122 - maybe useful to indicate to the reader that the Outcome measures are appearing later on

Page 7, Line 149-150 - were the intervention group not starting the intervention at the same time point, i.e. after cast removal? Please make very clear the time points for the intervention group, as this may impact on whether there were other parameters introduced (linking also to a statement made on page 11, Line 221).

Page 8, Line 164 - ideally, the ones that did not participate would also offer ideas as to why they have not.

Page 8, Line 174 and Page 9, Line 175 - please remove the duplication

Page 8, Line 175 - where is this analysis stemming from? No such 'plan' was described, nor justified by the Introduction. No discussion was present around sex- / gender-based differences. Further, no such analysis was conducted (or at the very least, presented).

Statistical analysis: The reason for the correlation testing should be included here. It exists in the Results but it should be here; to enable reduction of the outcome measures. There is also no description of how the qualitative data will be analysed.

Results: The qualitative results should be better analysed and described. Perhaps a helpful starting point that would help with all of the above is Anderson (https://www.ncbi.nlm.nih.gov/pmc/articles/PMC2987281/) and Braun and Clarke (https://www.tandfonline.com/doi/abs/10.1191/1478088706qp063oa).

Page 13, Line 265: Why were they not captured, the participants recorded health-related events? It also contradicts Page 15, Line 313.

Considerations for future trial: this should appear in the end, when all parameters have been presented to the reader.

Outcomes: The criterion for removing redundant outcome measures was an R>0.8 ; none of the proposed ones me that criterion.

Outcome measures: In the table, you should present the results at all time points, not only at baseline.

Page 14, Lines 305: how was this sample size calculated?

Page 15, Lines 313-316 - not sure of the purpose of this para; no adverse effect were observed (albeit earlier it was stated they were not formally captured), so why is the reference important?

Page 18, Line 379 - please replace 'it's not' with 'it is not'

Discussion: as per the main comment earlier, it needs clarification and better structure to convey the results, considering the actual aim of teh study ('so what') and the purpose of a feasibility study.

7. PLOS authors have the option to publish the peer review history of their article (what does this mean?). If published, this will include your full peer review and any attached files.

Reviewer #1: No

Reviewer #2: No

---

## [Author Response · Author response to Decision Letter 1]

3 Oct 2024

6. Review Comments to the Author

Reviewer #2: The authors have attempted to address my previous comments; this was successful at some points, while less so in others. I will try and discuss further the remaining points, with the aim to provide more clarity on the issues I think require attention. The more key ones are presented first.

Discrepancies throughout: there are still discrepancies in the manuscript. The selection of the age range is based on the thought processes, evidenced in the Introduction. To then recommend increasing the age range or stating that the 'age restriction.. reduced the.. participants', questions the rationale for those decisions. Further, increasing the age range potentially induces different issues (assuming an age-based digital divide, if the age range is increased to younger people, the difficulty in online assessment may be less, while if it increased to older people, the difficulty in online assessment may be more; transportation maybe less / more of a problem; time availability might be more / less of a problem (e.g. if >65 are retired) etc). Therefore, the solution can not be to alter parameters that would affect the process described.

This is a great point. I have removed the suggestion of increasing age range. 

Aim of the study: the aim of the programme is described as 'to complement rehabilitation of the DRF'. Its possible implementation is not clear. The description appears to present is as a standard addition to the usual care, while it appears it could be an option. In that respect, will it work for those who choose that option? Why is there insistence on a single site when travelling was reported as an issue, and delivering online at a different site might increase the uptake (close enough for the in-person visits, and online so they are reduced). There seems to be a need for a clearer 'so what' in the study.

I don’t believe there is insistence on a single site. I reported that we only collected from one site for this pilot study but suggested adding additional sites would help support reaching our sample size target for a fully powered RCT. I fully agree that adding more sites can increase the uptake as well. I’ll make sure that’s more clear in the discussion. 

Recommendations: There are several recommendations that seem unsupported and would alter the parameters of the feasibility trial. Please revisit and address accordingly. Perhaps a helpful starting point that would help with all of the above is Lancaster and Thabane (https://pilotfeasibilitystudies.biomedcentral.com/articles/10.1186/s40814-019-0499-1). This would help with providing a clearer direction.

Thank you for this recommendation. I reviewed the CONSORT reporting guidelines and reviewed the extension for non-randomized trials, despite this being a randomized trial. 

Specific comments

Page 5, Line 95: please replace 'is compliment rehabilitation' with 'is to complement rehabilitation'

Thank you, done. 

Page 5, Line 102: Please convert the question to statement.

Done. 

Page 6, Line 116 - the ratio has already been reported earlier

This has been removed. 

Page 6, Line 120 and Page 6, Line 128: Please keep one of the two statements re Ethics. Also add the Body that provided the Ethical approval

Thank you. Ethics approval is only stated once, but that it was approved by the Western Research Ethics Board. 

Page 6, Line 122 - maybe useful to indicate to the reader that the Outcome measures are appearing later on

Thank you, done. 

Page 7, Line 149-150 - were the intervention group not starting the intervention at the same time point, i.e. after cast removal? Please make very clear the time points for the intervention group, as this may impact on whether there were other parameters introduced (linking also to a statement made on page 11, Line 221).

Good point. I have added clarification that the intervention group started after their cast was removed. 

Page 8, Line 164 - ideally, the ones that did not participate would also offer ideas as to why they have not.

That’s a good point. Unfortunately we did not interview participants in the control group. 

Page 8, Line 174 and Page 9, Line 175 - please remove the duplication

Thank you. The a priori success criteria was removed, as it was all captured in the outcome measure section. 

Page 8, Line 175 - where is this analysis stemming from? No such 'plan' was described, nor justified by the Introduction. No discussion was present around sex- / gender-based differences. Further, no such analysis was conducted (or at the very least, presented).

Statistical analysis: The reason for the correlation testing should be included here. It exists in the Results but it should be here; to enable reduction of the outcome measures. There is also no description of how the qualitative data will be analysed.

Results: The qualitative results should be better analysed and described. Perhaps a helpful starting point that would help with all of the above is Anderson (https://www.ncbi.nlm.nih.gov/pmc/articles/PMC2987281/) and Braun and Clarke (https://www.tandfonline.com/doi/abs/10.1191/1478088706qp063oa).

Thank you for this feedback. I have addressed this in multiple places throughout the manuscript. 

Page 13, Line 265: Why were they not captured, the participants recorded health-related events? It also contradicts Page 15, Line 313.

Considerations for future trial: this should appear in the end, when all parameters have been presented to the reader.

This is a good point. I have removed that adverse events weren’t captured, as they were. 

Outcomes: The criterion for removing redundant outcome measures was an R>0.8 ; none of the proposed ones me that criterion.

Good point. I have clarified that 

Outcome measures: In the table, you should present the results at all time points, not only at baseline.

Thank you. The additional time points have been added. 

Page 14, Lines 305: how was this sample size calculated?

More of a description has been added. 

Page 15, Lines 313-316 - not sure of the purpose of this para; no adverse effect were observed (albeit earlier it was stated they were not formally captured), so why is the reference important?

This paragraph was deleted. 

Page 18, Line 379 - please replace 'it's not' with 'it is not'

Done. 

Discussion: as per the main comment earlier, it needs clarification and better structure to convey the results, considering the actual aim of teh study ('so what') and the purpose of a feasibility study.

Good point. I have tried to streamline the discussion to make it link more to the feasibility study and discussion the potential for the future trial.

---

## [Decision Letter · Decision Letter 2]

17 Oct 2024

Hands Up Program: results of a feasibility study of a randomized controlled trial of a bone health exercise and education program for adults aged 50-65 post distal radius fracture

PONE-D-24-19994R2

Dear Dr. Ziebart,

We’re pleased to inform you that your manuscript has been judged scientifically suitable for publication and will be formally accepted for publication once it meets all outstanding technical requirements.

Kind regards,

Enock Madalitso Chisati, PhD

Academic Editor

PLOS ONE

Additional Editor Comments (optional):

Reviewers' comments:

Reviewer's Responses to Questions

**Comments to the Author**

1. If the authors have adequately addressed your comments raised in a previous round of review and you feel that this manuscript is now acceptable for publication, you may indicate that here to bypass the “Comments to the Author” section, enter your conflict of interest statement in the “Confidential to Editor” section, and submit your "Accept" recommendation.

Reviewer #2: All comments have been addressed

2. Is the manuscript technically sound, and do the data support the conclusions?

Reviewer #2: Yes

3. Has the statistical analysis been performed appropriately and rigorously? 

Reviewer #2: Yes

4. Have the authors made all data underlying the findings in their manuscript fully available?

Reviewer #2: Yes

5. Is the manuscript presented in an intelligible fashion and written in standard English?

Reviewer #2: Yes

6. Review Comments to the Author

Reviewer #2: (No Response)

7. PLOS authors have the option to publish the peer review history of their article (what does this mean?). If published, this will include your full peer review and any attached files.

Reviewer #2: No

---

## [Editor Report · Acceptance letter]

25 Oct 2024

PONE-D-24-19994R2 

PLOS ONE

Dear Dr. Ziebart, 

I'm pleased to inform you that your manuscript has been deemed suitable for publication in PLOS ONE. Congratulations! Your manuscript is now being handed over to our production team.

Kind regards, 

on behalf of

Dr. Enock Madalitso Chisati 

Academic Editor

PLOS ONE